# Assessing Knowledge and Attitude of Healthcare Professionals on Biosimilars: A National Survey for Pharmacists and Physicians in Taiwan

**DOI:** 10.3390/healthcare9111600

**Published:** 2021-11-21

**Authors:** Samantha Yun-Kai Poon, Jason C. Hsu, Yu Ko, Shao-Chin Chiang

**Affiliations:** 1Department of Pharmacy, School of Pharmaceutical Sciences, National Yang Ming Chiao Tung University, Taipei 112, Taiwan; s.poon26@nycu.edu.tw; 2International Ph.D. Program in Biotech and Healthcare Management, College of Management, Medical University, Taipei 106, Taiwan; jasonhsu@tmu.edu.tw; 3Clinical Data Center, Office of Data Science, Taipei Medical University, Taipei 106, Taiwan; 4Research Center of Data Science on Healthcare Industry, College of Management, Taipei Medical University, Taipei 106, Taiwan; 5Department of Clinical Pharmacy, School of Pharmacy, College of Pharmacy, Taipei Medical University, Taipei 110, Taiwan; nancyko@tmu.edu.tw; 6Research Center of Pharmacoeconomics, College of Pharmacy, Taipei Medical University, Taipei 110, Taiwan; 7Department of Pharmacy, Koo Foundation Sun Yat-Sen Cancer Center, Taipei 112, Taiwan

**Keywords:** biosimilar, healthcare professional, survey questionnaire

## Abstract

Despite the first approval of biosimilars’ in 2010, biosimilar products usage has remained low in Taiwan. This cross-sectional survey study assessed healthcare professionals’ (HCPs)—hospital pharmacists, oncologists, and rheumatologists—knowledge, and attitudes toward biosimilars. More precisely, their knowledge and attitude towards biosimilars’ current usage and regulations in Taiwan were analyzed. The mean ± standard deviation knowledge score was 2.56 ± 0.86 out of 4 (*n* = 395), and a difference in knowledge score was determined according to the hospital types (*p* = 0.004). Rheumatologists possessed significantly higher confidence in their knowledge of biosimilars than other HCPs (*p* = 0.001). Pharmacists showed the highest acceptance—and rheumatologists the least—for switching patients from reference drugs to biosimilars (*p* = 0.02). Hospital type was associated with the respondent’s confidence in their knowledge (*p* = 0.04) and the preference for distinguishable naming of biosimilars (*p* = 0.007). Their knowledge scores were associated with their confidence in the efficacy and safety of biosimilars (*p* = 0.02). The study found that the current level of biosimilar knowledge of HCPs in Taiwan is low. The higher the knowledge score, the greater the confidence in biosimilars and the familiarity with relevant regulations.

## 1. Introduction

Biologics are essential for many severe chronic diseases, including cancer and autoimmune diseases. For public health insurance, making reimbursement decisions that consider the high costs of biologic drugs without compromising treatment quality is a common practice globally. With many high-priced blockbusters approaching patent expiration, the rise of biosimilars offers hope for a solution. The global biosimilar market has grown rapidly over the years. Experts estimated that the market will be worth more than USD 60 billion by 2026, up from 30 billion in 2020 [1,2,3,4]. In Taiwan, the National Health Insurance (NHI) reimbursement claim for biologics reached a total of TWD 30 billion (approx. USD 1.07 billion) in 2018, accounting for 16% of overall drug expenditures. However, the NHI reimbursement for biosimilars was estimated to be less than 1% of all biologics [5].

The World Health Organization (WHO) has defined a biosimilar as “a biotherapeutic product which is similar in terms of quality, safety, and efficacy to an already licensed reference biotherapeutic product” [6]. Biosimilars could positively impact the financial sustainability of the healthcare system through price competition and drive drug innovation. However, these benefits can be achieved only if biosimilars are widely used, and to be widely used in clinical settings, biosimilars must be accepted by both healthcare professionals (HCPs) and patients. In particular, the awareness and attitude of physicians and pharmacists toward biosimilars are the key determinants of clinical decisions to use them. For their successful adoption, it is urgently required to educate HCPs about biosimilars not only to increase access but also to improve knowledge of the innovation, quality, and value of biosimilars. A close partnership among all involved stakeholders in the healthcare system—from governmental agencies, policy-makers, and organizations to HCPs and patients—is essential to propagate information and knowledge on biosimilars [7,8].

The first biosimilar product, somatropin (Omnitrope^®^), was approved by the Taiwan Food and Drug Administration (FDA) in 2010. The first monoclonal antibody biosimilar product, infliximab, was approved by the Taiwan FDA in 2015. As of June 2021, 18 biosimilar products of nine active ingredients have been approved in Taiwan, of which 11 products (eight active ingredients) have received NHI coverage. Nevertheless, over the past five years, the use of biosimilar products has remained low in Taiwan. In the early stages of promoting biosimilars in various countries, it is typical for HCPs, especially physicians and pharmacists, to resist using them due to limited clinical experience and understanding [9,10,11,12,13]. The purpose of this study was to assess HCPs’ knowledge and attitude toward biosimilars during their early stage in Taiwan.

## 2. Materials and Methods

### 2.1. Study Design

Our study was a cross-sectional survey. We developed a 16-item questionnaire tailored to the current usage and regulations of biosimilars in Taiwan. The questionnaire consisted of three sections: (1) respondents’ demographic and background information; (2) respondents’ knowledge of biosimilars and their regulations (four multiple-choice questions); (3) respondents’ attitude toward issues related to biosimilars, including assessing their knowledge, being confident about biosimilars’ efficacy and safety, separating the name of a biosimilar product from that of its reference drug, switching a reference drug with a biosimilar product for patients, using biosimilar products only in patients who have never used the reference drug, and assessing the understanding of the Taiwan FDA regulations regarding biosimilar medications.

For the knowledge questions, each respondent was assessed on a point-based system of one to four depending on their number of correct answers. The respondent’s attitude was accessed using a 5-point Likert scale, scoring their level of agreement, with ‘1′ being ‘strongly disagree’ and ‘5′ being ‘strongly agree’. 

As part of the questionnaire design process, several revisions were made based on feedback from five experts in the field. The validity of all listed questions was rated from one to five points; only questions that received a full score were presented in the questionnaires. Moreover, to confirm that the respondents provided a well-thought-out response when filling out the survey, a reverse question was added to Section 3 to test the reliability of each respondent’s answers. 

### 2.2. Participants

We surveyed the HCPs in all medical centers and regional hospitals in Taiwan, which are first- and second-tier hospitals, according to the hospital accreditation conducted by the Taiwan Ministry of Health and Welfare, where biosimilars are mainly prescribed. Three groups of HCPs were recruited: rheumatologists, oncologists, and pharmacists. We aimed to recruit nine participants (three for each profession) from each medical center and six participants (two for each profession) from each regional hospital. For this study, we recruited hospital pharmacists in a decision-making position, from the chief pharmacist onwards. Subjects working in the following hospitals were excluded: district hospitals (i.e., third-tier hospitals), hospitals located in offshore islands in Taiwan, and hospitals specializing in obstetrics, pediatrics, or traditional Chinese medicines where the usage of biosimilars was expected to be low. Furthermore, pharmacists from the principal investigator’s hospital were not recruited because of their workplace relationship. 

### 2.3. Survey Administration

The data collection period was between 10 December 2020 and 3 March 2021. The survey questionnaires were delivered by mail to 19 medical centers and 76 regional hospitals. The HCPs in the three target professions were randomly selected and offered a gift card of TWD 200 as an incentive to participate. The surveyed HCPs were asked to complete and return the questionnaire anonymously.

### 2.4. Data Analysis

The responses to each survey question were summarized and analyzed using the IBM SPSS Statistics for Windows, version 24 (IBM Corp., Armonk, NY, USA). Categorical variables were presented as counts and percentages, whereas continuous variables were presented as means and standard deviations. Graphic visualization of data is represented by bar charts depicting the relative frequency of answer choices. The chi-square test was used to examine the relationships among respondents’ demographics, knowledge, and attitudes. Statistical significance was set at *p* < 0.05. Since the many categories of a Linkert scale could obscure the purpose of the response, the number of categories was reduced. Therefore, to improve the outcome of the analysis, ‘strongly disagree’ was merged with ‘disagree’ and ‘strongly agree’ with ‘agree’, leaving ‘neutral’ as it was to form three response categories. Furthermore, the correlation of knowledge scores with the respondents’ demographic characteristics and attitudes was analyzed by further categorizing the respondents into those who achieved a full score and those who did not.

## 3. Results

### 3.1. Demographics

A total of 408 questionnaires were returned (response rate = 65.3%), and 13 of them did not pass the reliability test. Therefore, 395 questionnaires were valid for the analysis and evaluation. 

Table 1 lists the breakdown of the participants’ demographics. Among the respondents, 164 (41.5%) were pharmacists, 123 (31.1%) were oncologists, and 108 (27.4%) were rheumatologists. The majority of the respondents were male (62.5%), 40–49 years old (44.3%), working in northern Taiwan (47.0%), and from regional hospitals (63.5%). 

### 3.2. Knowledge

We tested respondents with four multiple-choice questions related to biosimilars in Taiwan. The mean ± standard deviation (SD) total score was 2.56 ± 0.86 out of 4 points, with pharmacists, oncologists, and rheumatologists scoring 2.52 ± 0.90, 2.62 ± 0.81, and 2.56 ± 0.86, respectively.

The proportions of respondents who answered each question correctly, as well as the proportion who answered all questions correctly, are illustrated in Figure 1. Overall, over 86% of the respondents correctly answered the two questions regarding the definition of clinical trial requirements for biosimilars in Taiwan. However, only 56.9% of the respondents knew about the approved indications for biosimilars in Taiwan, and only 26.6% knew that the regulations require more physiochemical analysis and biological function data of biosimilars compared with their reference drugs. Moreover, only 56 respondents (14.2%) answered all four questions correctly. Among the three health professions, 15.3% of the pharmacists, 13.1% of the oncologists, and 13.9% of the rheumatologists received a perfect score; however, no difference in the proportions was found between the groups (*p* = 0.88, Table 2). Of the five demographic characteristics (gender, region, profession, age groups, and hospital type), only hospital type showed a significant association with the respondents’ knowledge scores. A greater proportion of respondents from medical centers (20.8%) had a perfect score than those from regional hospitals (10.4%) (*p* = 0.004).

Q6. Which of the following is the correct definition of biosimilars?Q7. Which of the following regarding the clinical trial requirements for biosimilars in Taiwan is correct?Q8. Which is the correct statement regarding approved indications for biosimilars in Taiwan?Q9. Which of the following best describes the physiochemical/functional analysis of biosimilars required by regulations? 

**Table 2 healthcare-09-01600-t002:** The association between respondents’ knowledge and the five demographic characteristics.

	Full Score *n* (%)	95% CI	*p*-Value
Specialty			0.88
Pharmacy	25 (15.2)	(9.7–20.7)	
Oncology	16 (13.1)	(7.1–20.7)	
Rheumatology	15 (13.9)	(7.4–20.4)	
Gender			0.36
Male	38 (15.4)	(10.9–20.0)	
Female	17 (12.1)	(6.7–17.4)	
Age group			0.82
20–39	14 (15.6)	(8.1–23.0)	
40–49	23 (13.2)	(8.1–18.2)	
50–59	15 (16.1)	)8.7–23.6)	
60+	4 (10.8)	(0.8–20.8)	
Hospital types			0.004 *
Medical center	30 (20.8)	(14.2–27.5)	
Regional hospital	26 (10.4)	[6.6–14.1]	
Regions			0.21
North	29 (15.5)	(10.5–21.0)	
Central	8 (8.4)	(2.8–14.0)	
South	15 (15.5)	(0.8–22.7)	

Note: The number of respondents with a full score for the knowledge questions n (%). Gender [*n* = 387, Male (246), Female (141)], “prefer not to say” was excluded; Regions [*n* = 376, North (184), Central (95), South (97)], “Eastern region” was excluded. * *p* < 0.05.

### 3.3. Attitudes

Seven questions on a 5-point Likert scale were used to assess respondents’ attitudes toward biosimilars (Figure 2). Overall, 62.8% of the respondents reported that they had a good understanding of biosimilars. Additionally, it was found that a higher proportion of rheumatologists (75.0%) felt they had such an understanding compared to those in the other two health professions (*p* = 0.001, Appendix A). Moreover, more male respondents (male 69.2% vs. female 53.2%, *p* = 0.007) and more respondents from medical centers (medical centers 70.8% vs. regional hospitals 58.2%, *p* = 0.04) believed that they were knowledgeable about biosimilars. 

Q10. I possess a good understanding of biosimilar products.Q11. Generally, I feel comfortable prescribing biosimilars to patients because I am confident about their safety and efficacy.Q12. The nonproprietary name of biosimilars should be distinguishable from the reference product. Q13. If necessary, I accept the switch to a biosimilar product for patients receiving treatment of its reference product.Q14. I would only prescribe a biosimilar product to those who have never received treatment with its reference drug (i.e., biologic-naïve).Q15. I am familiar with the regulations of the Taiwan Food and Drug Administration (TFDA) on biosimilars. Q16. The non-proprietary name of biosimilars should not be distinguishable from the reference product (respondent reliability test).

The majority of the respondents agreed (>80%) that the nonproprietary name of a biosimilar and that of the reference drug should be distinguishable. Additionally, a higher proportion of respondents from medical centers (94.4%) agreed on this compared to those from regional hospitals (83.6%) (*p* = 0.007). 

Overall, around half the respondents felt comfortable with patients’ use of biosimilars because they had confidence in their safety and efficacy. However, only one-third of the respondents agreed that biosimilar products should only be used by patients who have never received their reference drugs (i.e., biologic-naïve). No difference was found in attitudes toward these two issues among any demographic subgroups.

Although approximately half of the respondents (45.8%) agreed to switch a patient receiving a reference drug to a biosimilar, the respondents’ attitudes varied among health professions (*p* = 0.02). Specifically, over half of the pharmacists agreed with the switch compared to only around one-third of the rheumatologists. 

### 3.4. Knowledge and Attitudes

The respondents’ self-reported confidence in the safety and efficacy of biosimilars was positively associated with their total knowledge score of biosimilars (*p* = 0.02) (Table 3). The majority of the respondents (67.9%) who obtained a perfect knowledge score expressed confidence in biosimilars compared to only 16.1% of those who did not. There was also an association between respondents’ self-reported familiarity with the Taiwan FDA’s regulations for biosimilars and their total knowledge score (*p* = 0.01); those who reported being familiar with the Taiwan FDA’s regulations were more likely to receive a perfect knowledge score. 

## 4. Discussion

To the best of our knowledge, this is the first nationwide questionnaire study on the relationship between knowledge and attitude among HCPs in Taiwan, particularly rheumatologists, oncologists, and pharmacists. Our investigation suggests that knowledge of biosimilars is still relatively low among HCPs at this early stage of use in Taiwan. From our findings, it is clear that the better the biosimilar knowledge of the respondents, the higher their confidence in biosimilar products.

Only 14.2% of the respondents obtained a full score (4 points), and the mean total score was 2.56 ± 0.86 points, which indicates a low level of understanding of biosimilars. There was no difference in knowledge scores among the three health professions. However, one-fold more respondents from medical centers scored 100% than those from regional hospitals. This result may indicate that HCPs from regional hospitals may require more extensive education.

In terms of attitude, the following correlations showed statistically significant differences: the highest percentage of rheumatologists expressed being knowledgeable about biosimilars, followed by oncologists and pharmacists. Interestingly, the rheumatologists showed the least confidence in the safety and efficacy of biosimilars. Compared to oncologists, rheumatologists also showed a greater reluctance to switch. A total of 70.8% of the respondents from medical centers and 58.2% from regional hospitals felt knowledgeable about biosimilars. This result is consistent with our findings related to the objective knowledge score and the type of hospital accreditation. We also observed pharmacists’ higher acceptance of biosimilars use than the other two professions. Additionally, we found ‘hospital types’ to be a potential predictor variable for this study, as it was the only demographic characteristic observed with a significant correlation with the respondents’ knowledge score, subjective confidence in their knowledge, and the wish to distinguish biosimilar naming.

Several studies reported that during the early promotion stage, most physicians prefer to prescribe biosimilars to biologic treatment-naïve patients and become more concerned while considering the switch from a reference drug to a biosimilar [12,13,14,15,16,17,18,19,20,21,22]. Our investigation shows the opposite in Taiwan, with a higher percentage of oncologists and pharmacists agreeing with switching than rheumatologists, and this finding aligns well with a few other studies focusing on rheumatologists [9,15,23].

Studies from different countries in their early promotion stage suggested that most HCPs, particularly physicians, had incomplete or only basic knowledge regarding biosimilar products and their benefits [9,11,15,16,17,24,25]. Conversely, pharmacists possess greater familiarity with biosimilars compared to physicians [16,17,20]. Moreover, opinions on biosimilars were divided among physicians from different professions, and rheumatologists tended to have less favorable opinions on biosimilars [15,16,23,24,26].

In addition to this study, only two other ones in this area of study are from Asian countries. A 2015 Japanese study conducted during the introductory stage of biosimilars stated a low awareness and a deep concern for biosimilars among physicians. The other is a recent study from South Korea reporting a positive opinion on biosimilars among physicians after a rapid expansion of the biosimilar market and domestic development starting in 2012. However, recently they became less confident regarding switching from the reference drug to a biosimilar, and about 30% of physicians stated a low willingness to switch. The authors also reported that the willingness to prescribe biosimilars increases with the physician’s medical experience and surgical oncologists had the most positive attitude toward biosimilars among physicians [27].

Follow-up studies on changes in attitude and knowledge are scarce. The European Crohn’s and Colitis Organisation (ECCO)’s follow-up survey published in 2016 reported improved biosimilar knowledge among European gastroenterologists from their investigation in 2013. Attitudes among ECCO members regarding biosimilars were considered “conservative” in 2013. Two years later, they noted a drastic change: opinions on biosimilars had improved to a favorable and confident position. The authors concluded that extensive education at the postgraduate level, published evidence, and increased utilization of biosimilars across the EU countries, all affected this change [18,19]. Furthermore, van Overbeeke et al. also suggested a positive association between biosimilar education and biosimilar knowledge [14]. However, several European countries still reported low to moderate degrees of knowledge, awareness, and trust in biosimilars [12,13].

In our study, the rheumatologists had the highest percentage among the three HCPs in expressing their confidence in the knowledge of biosimilars; they also expressed the lowest trust in the efficacy and safety of biosimilars. This is probably because they receive adequate opportunities for biosimilar education, but the educators are not from a single authoritative source [11]. The diverse responses related to acceptance and knowledge of biosimilars among HCPs emphasize the necessity for providing continuous and positive evidence-based information and education. Their knowledge would subsequently be transferred to the patients and mitigate their concerns. European researchers have noted that passive education (i.e., website information) has limited efficiency. Dynamic delivery of tailored messages and information targeting different HCP groups and therapeutic areas ensures effective education. Successful delivery of information and education depends on collaborative efforts between government agencies and different stakeholders [12]. To further elaborate, the lack of incentive policies and promotion in Taiwan contributed to the low uptake of biosimilars; this led to low acceptance among HCPs and knowledge gaps due to unfamiliarity. European researchers concluded that physicians informed through official and reliable sources tend to view biosimilars positively and are more willing to improve their knowledge about biosimilars. Likewise, those who partook in a shared savings system with better-aligned reimbursements were more likely to adopt biosimilars and change their prescription patterns [28]. Hence, recognizing their effort through incentives and other forms of shared benefits encourages long-term commitments. Through increasing reimbursement for providers, biosimilar uptake can be improved and potentially drive savings for the healthcare system [13,29]. At the current state in Taiwan, biosimilars undergo a hospital listing process akin to novel drugs. The delay in access and not having a common goal among stakeholders mean that this will be a long-term battle for Taiwan. Nonetheless, with time we are hopeful to see the progressive changes biosimilars may bring to Taiwan’s healthcare system.

### Limitations and Future Perspectives

This study used only four questions to evaluate the level of knowledge of HCPs. The four questions cover the definition, requirement of clinical trials, approval of indications, and relative portions of the totality of the evidence. There are more aspects of regulations related to biosimilars, including the format of the package insert, the requirement of indication extrapolation, and the demand for pharmacovigilance. Therefore, the full score may not represent a full range of knowledge of biosimilars. Medical professions other than rheumatologists, such as gastroenterologists and dermatologists, may also prescribe monoclonal antibody biosimilars for autoimmune diseases. However, the knowledge and attitudes of these HCPs were not presented in this study. Furthermore, to maintain the anonymity of the survey responses, instead of giving their specific age, the participants were asked to select an age range. This had its limitations, as we found that the 40–49 age group contributed to >40% of the sample size and also restricted the types of analysis we could perform. In the future, we would like to conduct a follow-up study when biosimilar use reaches a relatively mature status in Taiwan.

## 5. Conclusions

The current level of knowledge regarding biosimilars among HCPs is generally low during this introductory phase in Taiwan. Even when their knowledge—and self-awareness of this knowledge—is high, it does not always translate into greater confidence in the safety and efficacy of biosimilar products.

## Figures and Tables

**Figure 1 healthcare-09-01600-f001:**
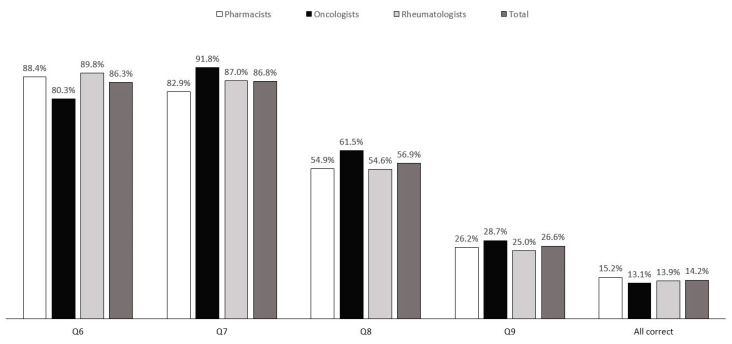
The proportion of respondents (%) who correctly answered knowledge questions (Q6–9).

**Figure 2 healthcare-09-01600-f002:**
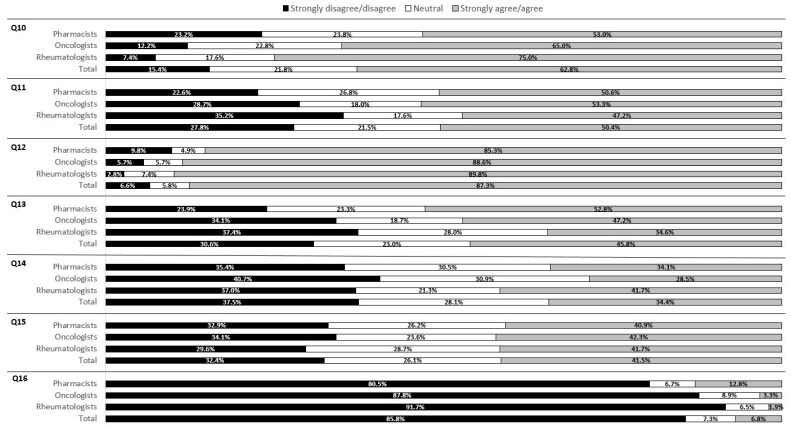
Respondents’ attitudes toward biosimilars by professions.

**Table 1 healthcare-09-01600-t001:** Demographic characteristics of respondents (*n* = 395).

Characteristics	Pharmacists *n* = 164 (41.5%)		Oncologists *n* = 123 (31.1%)		Rheumatologists *n* = 108 (27.4%)		Total		Total [95% CI]
	n	%	n	%	n	%	n	%	
Gender									
Male	65	39.6	104	84.6	78	72.2	247	62.5	57.7–67.2
Female	95	58.0	17	13.8	29	26.9	141	35.7	31.0–40.4
Prefer not to say	4	2.4	2	1.6	1	0.9	7	1.8	0.5–3.1
Age (years)									
20–39	38	23.2	29	23.6	23	21.3	90	22.8	18.7–26.9
40–49	73	44.5	57	46.3	45	41.7	175	44.3	39.4–49.2
50–59	47	28.6	20	16.3	26	24	93	23.5	19.3–27.7
60+	6	3.7	17	13.8	14	13	37	9.4	6.5–12.3
Hospital type									
Medical center	54	32.9	47	38.2	43	39.8	144	36.5	31.8–41.2
Regional hospital	110	67.1	76	61.8	65	60.2	251	63.5	58.8–68.2
Region									
North	81	49.4	56	45.5	47	43.5	184	47.0	41.7–51.5
Central	40	24.4	26	21.2	29	26.9	91	23.0	18.8–27.2
South	35	21.3	34	27.6	29	26.9	98	25.0	20.5–29.1
East	8	4.9	7	5.7	3	2.7	18	5.0	2.5–6.7

**Table 3 healthcare-09-01600-t003:** The association between attitude statements and respondents’ knowledge.

Attitude Statements	*n* (% within Respondents with Full Score)			*p*-Value
	Disagree	Neutral	Agree	
Q10. I possess a good understanding of biosimilar products.	4 (7.1)	9 (16.1)	43 (76.8)	0.05
Q11. Generally, I feel comfortable prescribing biosimilars to patients because I am confident about their safety and efficacy.	9 (16.1)	9 (16.1)	38 (67.8)	0.02 *
Q12. The nonproprietary name of biosimilars should be distinguishable from the reference product.	7 (12.5)	1 (1.8)	48 (85.7)	0.07
Q13. If necessary, I accept the switch to a biosimilar product for patients receiving treatment of its reference product.	16 (28.6)	10 (17.9)	30 (53.5)	0.42
Q14. I would only prescribe a biosimilar product to those who have never received treatment with its reference drug (i.e., biologic-naïve).	22 (39.3)	12 (21.4)	22 (39.3)	0.48
Q15. I am familiar with the regulations of the Taiwan Food and Drug Administration (TFDA) on biosimilars.	11 (19.6)	12 (21.4)	33 (58.9)	0.01 *
Q16. The non-proprietary name of biosimilar drugs should not be distinguishable from the reference product (respondent reliability).	46 (82.1)	3 (5.4)	7(12.5)	0.17

Note: n % was represented in percentage within all respondents who achieved a full score (*n* = 56). * *p* < 0.05.

## Data Availability

The data presented in this study are available in the article and as Appendix A.

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
