# Peer review of "Assessing Knowledge and Attitude of Healthcare Professionals on Biosimilars: A National Survey for Pharmacists and Physicians in Taiwan"

_healthcare, 2021, doi:10.3390/healthcare9111600_

Round 1

Reviewer 1 Report

Topic is not novel but maybe good for the country's perspective (as the authors stated)

Intro and discussion acceptable BUT

Methodology should be discussed in further details WHILE

results are presented but are NOT SATISFACTORY, all of the data should be shared/presented within the manuscript.

Data analysis is not rigorous, just descriptive, it MUST be supported by a rigor inferential analysis. And if possible predictors could be determined as well.

Author Response

Thank you for taking the time to review our manuscript, please see the attachment for our responses. 

Reviewer 2 Report

While potentially of value to guide continuing medical education and even for pharma marketing, the analysis can dive deeper into further analysis to make the data collected more meaningful.

Major:

Firstly there is discrepancy between the number of participants which are not clear in abstract and in the results. - it said 408 questionnaires are returned, but when I add up the Male Female and Prefer not to say, I do not even get 200.

Stats need more work - no confidence interval, no test for normality, graphs no standard errors or deviations.

Not much meaning without segregating the respective specializations by their seniority within the specialization and other more in-depth analysis.

For a nationwide analysis, 95+65+4 is pretty low number. It should be discussed from the registry, what percentage this actually represents to know the level of extrapolatable data.

Discussion could benefit how to address the lack of knowledge in specific seniority grades as well as the specialization, and whether it makes sense and perhaps a bit of background on the treatment options - if for example referral to internal medicine is done after surgery or as a substitute - then this changes drug selection as well, and of course, something to be discussed with relevance even in brief is the approval and ease of biosimilar entry or those locally produced, no point knowing about biosimilars if they are not approved for prescription.

Author Response

(The authors gave the same response as above.)

Reviewer 3 Report

Authors assessed knowledge and attitudes toward biosimilars by a survey study involving Health Care Professionals, hospital pharmacists, oncologists, and rheumatologists.

At this aim Authors used a questionnaire which explored respondents’ knowledge of biosimilars and their regulations (four multiple-choice questions) and respondents’ attitude toward issues related to biosimilars. Moreover a 5-point Likert scale was used to answer the attitude questions.

What they found is that the current level of knowledge regarding biosimilars among HCPs is generally low, with a low confidence in the 125 safety and efficacy of biosimilar products.

The topic is interesting, even if it is particularly important for the local real life of Taiwan.

Minor point

I suggest Authors explain in Methods the score system they used for the statistical evaluation of questionnaire.

Author Response

(The authors gave the same response as above.)

Round 2

Reviewer 2 Report

Comments are addressed. Proofreading will improve the manuscript.